# Venetoclax Plus CyBorD Induction Therapy and Venetoclax Maintenance Treatment for Immunoglobulin Light Chain Amyloidosis with t(11;14) Translocation

**DOI:** 10.3390/curroncol32020063

**Published:** 2025-01-26

**Authors:** Gréta Garami, Omar Obajed Al-Ali, István Virga, Anita Gulyás, Judit Bedekovics, István Tornai, Árpád Illés, Ferenc Magyari

**Affiliations:** 1Division of Hematology, Department of Internal Medicine, Faculty of Medicine, University of Debrecen, 4032 Debrecen, Hungary; obajed.omar@med.unideb.hu (O.O.A.-A.); virga.istvan@med.unideb.hu (I.V.); gulyas.anita@med.unideb.hu (A.G.); illes.arpad@med.unideb.hu (Á.I.); magyari.ferenc@med.unideb.hu (F.M.); 2Doctoral School of Clinical Medicine, University of Debrecen, 4032 Debrecen, Hungary; 3Department of Pathology, Faculty of Medicine, University of Debrecen, 4032 Debrecen, Hungary; bedekovics.judit@med.unideb.hu; 4Division of Gastroenterology, Department of Internal Medicine, Faculty of Medicine, University of Debrecen, 4032 Debrecen, Hungary; itornai@med.unideb.hu

**Keywords:** immunoglobulin-related/light chain amyloidosis, plasma cell disorder, t(11;14) translocation, CyBorD, venetoclax, autologous stem cell transplantation, maintenance treatment, dFLC

## Abstract

**Background**: A total of 50% of patients with AL amyloidosis have t(11;14) translocation, allowing us to use the selective oral BCL-2 inhibitor venetoclax in their treatment. **Case presentation**: Our patient was admitted to the gastroenterology department due to weight loss and abdominal pain. An abdominal CT scan revealed some enlarged lymph nodes; therefore, he was referred to the hematology department. A bone marrow biopsy showed massive amorphous amyloid deposition. The sample was positive on Congo red staining and exhibited double refraction under a polarized light microscope. Serum-free light chains and the difference between involved and uninvolved free light chains (dFLCs) were elevated. Using fluorescent in situ hybridization, we detected t(11;14) translocation. Further examinations confirmed the involvement of the liver, colon and heart. Stage II AL amyloidosis was confirmed. Our patient received combined induction therapy with CyBorD and venetoclax due to the presence of the t(11;14) translocation. After six cycles, the patient achieved complete remission. Autologous stem cell transplantation (ASCT) was performed. At 100 days post-ASCT, the patient had complete hematologic remission. Venetoclax maintenance treatment was initiated. The follow-up examinations showed that the patient is in very good partial remission. **Conclusions**: In the case of our AL amyloidosis patient with t(11;14) translocation, the combined treatment with CyBorD and venetoclax was well tolerated and effective.

## 1. Introduction

Immunoglobulin-related/light chain amyloidosis (AL amyloidosis) or primary amyloidosis is the most common type of a rare systemic disease known as amyloidosis. According to the 2022 World Health Organization (WHO) classification, this disease falls into the plasma cell neoplasms and related disorders category, specifically within the subgroup of diseases with monoclonal immunoglobulin deposition, such as immunoglobulin-related amyloidosis [1]. The International Consensus Classification (ICC) is also used to classify lymphoid neoplasms. In this system, this disease is placed in the same subgroup as in the WHO classification but is referred to as immunoglobulin light chain amyloidosis (AL amyloidosis) [2]. In AL amyloidosis, the amyloidogenic proteins are monoclonal immunoglobulin light chains (kappa, lambda or both types and sometimes just their fragments) produced by plasma cells.

The symptoms of the disease are heterogeneous due to its multi-organ manifestations. The most commonly affected organs are the liver, kidneys, gastrointestinal tract and heart [3]. In terms of life expectancy, the cardiac manifestation of amyloidosis can lead to early death in patients [4].

The diagnosis is based on a microscopic evaluation of tissue samples of affected organs that are stained with Congo red and further immunohistochemical (IHC) examinations to clarify the type of amyloid light chain that is involved. It is important to note that the sensitivity of Congo red staining is highly dependent on the origin of the sample; for example, for bone marrow samples, it is lower than for samples from the liver, kidneys or heart muscle. Therefore, if there is a strong clinical suspicion and the staining is negative, it is recommended to follow up on the suspicion with other diagnostic methods, such as IHC [5]. Fluorescence in situ hybridization (FISH) is also important in the diagnostic procedure, as around 50% of patients with AL amyloidosis have t(11;14) translocation [6]. This chromosomal translocation results in the overexpression of cyclin D1 and B-cell lymphoma 2 (BCL-2) proteins in plasma cells, enabling them to avoid apoptosis [7]. The diagnosis protocol for AL amyloidosis is presented in Figure 1.

A curative treatment of AL amyloidosis has still not been developed; however, all patients require therapy to prevent further deposition of amyloid in the affected organs. Venetoclax is a selective oral inhibitor of the anti-apoptotic protein BCL-2. By inhibiting BCL-2, this drug promotes programmed cell death, leading to the natural destruction of plasma cells that produce abnormal proteins [7]. Since the first report of using a venetoclax-based regimen in t(11;14) AL amyloidosis, multiple case reports have demonstrated the efficacy of venetoclax for this indication [8,9].

Herein, we present the case history of one of our patients with t(11;14) immunoglobulin-related amyloidosis who was effectively treated with a combination of venetoclax, cyclophosphamide, bortezomib and dexamethasone.

## 2. Case Report

The examination of a 61-year-old male patient started in September 2022 at the Gastroenterology Department of the University of Debrecen due to weight loss and abdominal pain under the right costal margin. His medical history included hypertension, duodenal ulcer, gastritis and reflux disease, and his Eastern Cooperative Oncology Group Performance Status was classified as ECOG PS 1. Laboratory tests and imaging studies (colonoscopy, gastroscopy) revealed cholestasis, fatty liver disease and cholecystolithiasis. A shear wave elastography showed cirrhosis (87 kPa; stage F4; IQR: 6.5 kPa). An endoscopic retrograde cholangiopancreatography (ERCP) detected stenosis of the papilla of Vater, and bile duct lavage was performed, resulting in the initiation of bile drainage.

The examinations aimed at detecting malignancy were negative; however, the patient’s CEA and CA 19-9 values were elevated. His personal and family histories were both negative for malignancy.

In October 2022, abdominal computer tomography (CT) showed mediastinal, hilar and abdominal lymphadenopathy. The patient was referred to our clinic on 24 October 2022 due to B symptoms and lymphadenopathy. A bone marrow biopsy (posterior iliac crest) showed massive amorphous interstitial deposition (60% of the parenchyma), which was consistent with amyloidosis, without significant plasma cell infiltration (4–5% CD138 positivity with immunohistochemical staining). The sample that was faintly positive for Congo red exhibited apple-green birefringence under a polarized light microscope. While the serum protein electrophoresis did not detect monoclonal gammopathy (M component), the free light chain lambda and kappa were 74.8 mg/L and 9.46 mg/L, respectively, with a ratio (serum free lambda/kappa light chain) of 7.9. The difference between involved and uninvolved free light chains (dFLC) was 65.34 mg/L. At the same time, biopsies from the liver and colon were also taken. The liver sample showed amorphous, faint eosinophilic deposition and apple-green birefringence with Congo red staining under a polarized light. The tissue sample from the colon was negative at this time, but another bone marrow biopsy that we performed in December clearly confirmed the diagnosis of amyloidosis (Figure 2). We also found that 80% of the parenchyma was filled with the same homogenous deposition mentioned above.

The patient’s recent serum protein electrophoresis did not show monoclonal gammopathy (M component). A free light chain assay detected lambda and kappa concentrations of 80.1 mg/L and 10.2 mg/L, respectively, while the lambda/kappa ratio was 7.85, and the dFLC was 69.9. The immunohistochemistry (IHC) revealed strong CD138 and lambda and weak kappa and transthyretin positivity of the amyloid plaques, and the samples were also MUM 1 (multiple myeloma oncogene 1)- and cyclin D1-positive. These results urged us to repeat the first colon biopsy IHC. This time, a small amount of amyloid deposition was found with the Congo red staining. On the lymphoid gene panel, no abnormality was detected. Positron emission tomography–computed tomography (PET-CT) showed lymph nodes that were unsuitable for sampling with a low standardized uptake value (SUV_max_ was 1.8, with a reference liver SUV_max_ of 2.8).

We referred our patient for cardiac evaluation to assess any possible heart involvement. A transthoracic echocardiography (TTE) examination was performed, showing diastolic dysfunction (Stage I) and left ventricular enlargement.

The cardiac muscles of the interventricular septum and posterior wall were both 14–15 mm thick; the level of serum cardiac troponin T (cTnT) was 40.85 ng/L (normal range: <30 ng/L); and the level of the serum N-terminal portion of pro-brain natriuretic peptide type B (NT-proBNP) was exactly 409 ng/L (normal range: <176.8 ng/L). Concurrently, we performed an fluorescent in situ hybridization (FISH) assay to check the presence of t(11;14), the most common cytogenetic abnormality, which is also a prognostic and predictive biomarker for AL amyloidosis. The presence of this translocation is associated with more pronounced cardiac involvement, a lower hematologic response and reduced survival rates. The t(11;14) translocation was confirmed by means of FISH in our patient. Additionally, gene rearrangement in the immunoglobulin VH-VJ regions was also detected.

Based on results of the above-described evaluation and as per the Mayo 2012 criteria, the diagnosis of stage II immunoglobulin-related amyloidosis could be established.

The 28-day CyBorD regimen was initiated on 27 December 2022. This combination chemotherapy included cyclophosphamide (300 mg per day on days 1, 8 and 15), bortezomib (2.2 mg per day, administered subcutaneously on days 1, 4, 8 and 11) and dexamethasone (20 mg on days 1–2, 8–9, 15–16 and 22–23). Due to t(11;14), venetoclax was also administered from day 1 of the second CyBorD cycle. On a course of a venetoclax dose ramp-up at a dose level of 200 mg per day, the patient developed grade II thrombocytopenia; subsequently, 100 mg of venetoclax per day was administered throughout the induction and maintenance. During the treatment, we did not observe any other hematological toxicities or side effects necessitating intervention.

In AL amyloidosis, venetoclax is an off-label therapy. Regulatory permission to use venetoclax for this indication was granted by the National Institute of Pharmacy and Nutrition, and the treatment costs were covered by the Hungarian National Health Insurance Fund following a named-patient-based reimbursement application.

Shortly after treatment initiation, our patient’s clinical condition improved significantly. Furthermore, liver function tests also showed positive changes after the first cycle of therapy, inasmuch as the amount of GGT was reduced to a quarter and ALP decreased by almost half (Figure 3).

Shear wave elastography was repeated in February 2023, but unfortunately, no improvement was observed (139.4 kPa; stage F4). After the third treatment cycle, a control iliac crest bone marrow biopsy was performed. Despite clinical improvement, no significant quantitative or qualitative changes were observed. However, the concurrent serum electrophoresis showed the normalization of the patient’s free light chain levels (serum lambda light chain: 22.4 mg/L; kappa light chain: 9.53 mg/L; lambda/kappa light chain ratio: 2.35; dFLC: 12.87), and no M component was detected.

Ursodeoxycholic acid was also initiated due to significant liver involvement, signified by elevated serum alkaline phosphatase and γ-glutamyl transferase levels.

Between the 27 December 2022 and 23 May 2023, the patient received six cycles of CyBorD and venetoclax. As a result, complete remission was achieved by April 2023. A control echocardiography performed in June 2023 indicated some improvement, as the thickness of the left atrium wall had decreased from 47 mm to 38 mm. At the same time, the patient’s NT-proBNP levels had decreased to 204.2 ng/L, and his cTnT levels were measured at 18.47 ng/L. The follow-up bone marrow biopsies showed less than 5% of the initial clonal plasma cells. His serum-free light chain levels decreased by nearly 90%, the lambda/kappa ratio was 1.85, and the dFLC decreased to 5.33. Cholestasis tests also showed improvement (Figure 3 and Figure 4). With the patient reaching ECOG PS 0 and being in a sufficiently state deep of remission by April 2023, autologous stem cell transplantation (ASCT) has become a viable step forward. Autologous stem cell transplantation is considered highly effective in eligible patients, especially with pretransplant CyBorD induction followed by melphalan conditioning. A single granulocyte colony-stimulating factor (G-CSF) was used for stem cell mobilization. The collection of CD34-positive stem cells from the peripheral blood was successfully performed between the 2 and 4 June, during which we collected 4.62 × 10^6^ per kg of body weight without any complications. The conditioning regimen included 350 mg of melphalan (200 mg/m^2)^, which was followed by the reinfusion of 2.8 × 10^6^ per kg of body weight of cryopreserved stem cells on the 30 June.

A bone marrow sample was collected on the day 100 check-up on the 28 September from the iliac crest: the amorphous, homogeneous, pale eosinophilic deposit was still occupying a significant portion of the parenchyma (50%), although its quantity had somewhat decreased. This time, a monoclonal immunoglobulin G (IgG) lambda component was detectable in the sample (1.2 g/L). The levels of serum lambda and kappa free light chains had improved, and their difference had further decreased to 5.11 (Figure 4).

With his disease in hematological remission, the patient’s clinical status was excellent. A maintenance treatment with venetoclax at a dose of 100 mg per day was initiated, which he tolerated well without any hematological toxicities or major infective episodes.

We continue to see our patient for regular follow-up visits. Liver function tests (GGT and ALP) show further improvement (Figure 2). At the last check-up, he was in a good clinical condition, and his disease was in very good partial remission. The last serum protein electrophoresis, performed on 29 August 2024, detected an oligoclonal lambda component in the gamma region and revealed the following serum-free light values: lambda 32.1 mg/L; kappa 10.4 mg/L; and dFLC of 21.7 mg/L. The lambda/kappa ratio was 3.08.

## 3. Conclusions

Immunoglobulin-related or light chain amyloidosis (AL amyloidosis) is a rare and potentially life-threatening disorder caused by the deposition of abnormal light chain proteins in the tissues of various organs, leading to organ dysfunction and damage. Current therapies focus on eliminating the plasma cell clone that produces the amyloidogenic protein, thus preventing the formation of additional amyloid deposits and further organ damage [10].

Treatment guidelines, such as those provided by the International Society of Amyloidosis (ISA) and the European Hematology Association (EHA), can also be helpful in special cases.

Most chemotherapy regimens that are used for AL amyloidosis are extrapolated from multiple myeloma (MM), which is justified by the similarities between these diseases [11]. A non-exhaustive list of agents that are used includes corticosteroids (dexamethasone), alkylating agents (cyclophosphamide), immunomodulatory drugs (thalidomide), proteasome inhibitors (bortezomib) and daratumumab [12]. With the combination of these drugs, the hematologic response rate achieved is in the range of 60–94% [9].

Approximately 50% of patients with light chain amyloidosis have t(11;14), with a consequential overexpression of cyclin D1 and the B-cell lymphoma 2 (BCL-2) protein in plasma cells, allowing them to avoid apoptosis [7]. Venetoclax is a BH3 mimetic that selectively inhibits the anti-apoptotic protein BCL-2. By means of BCL-2 inhibition, this drug promotes programmed cell death in susceptible cells via intrinsic apoptosis, a process during which plasma cells produce abnormal proteins [7]. T(11;14) is also present in MM, where venetoclax is already recognized as an effective (although not registered) treatment option for patients with this cytogenetic abnormality and is used in monotherapy or combinations [13,14]. Most of the time, venetoclax proves to be an effective salvage therapy, with relapsed or refractory disease patients reaching complete remission (CR) or at least a very good partial remission (VGPR).

Results from the randomized double-blind multicenter Phase 3 BELLINI study raised numerous questions regarding the safety of venetoclax. Later, it was discovered that the higher mortality rate associated with venetoclax treatment was due to the very high efficacy of this targeted therapy. The inhibition of BCL-2 occurs not only in abnormal plasma cells but also in healthy ones. Severe hypogammaglobulinemia can develop, which favors the spread of the infectious agents [11].

Considering the presence of t(11;14) in both MM and AL amyloidosis and the key role of plasma cells in both diseases, venetoclax could be as useful in AL amyloidosis as it is in MM for patients who have this cytogenetic abnormality [11]. The first report of the activity of a venetoclax-based combination treatment (venetoclax, bortezomib, dexamethasone) in AL amyloidosis patients with a partial hematologic response to CyBorD was published by Leung et al. [8,9]. Since then, several new reports have emerged on using this salvage treatment, not only in patients whose response plateaued on CyBorD or those who were relapsed and refractory, but also in patients with severe cardiac or kidney involvement. These patients achieved CR or VGPR with venetoclax as a second-line therapy [15].

In a Multicenter International Retrospective Real-World Study, the combination of venetoclax with daratumumab or dexamethasone did not result in more favorable outcomes compared to venetoclax monotherapy [16]. On the other hand, the combination of daratumumab with CyBorD demonstrated positive results in the phase III ANDROMEDA trial, indicating its effectiveness for patients with AL amyloidosis [17].

As we have already established, venetoclax is most often used as a second-line or salvage therapy; however, we have less knowledge about its effectiveness when used in the early stages of these diseases [18]. Unfortunately, there are only scarce reports with limited value on venetoclax as a maintenance treatment for immunoglobulin-related amyloidosis. The maintenance treatment of AL amyloidosis is not a well-studied area, and there are no firmly established evidence-based treatment options in clinical practice yet [19]. There are two phase 2 trials on consolidation therapy post-autologous stem cell transplantation in patients with light chain amyloidosis who did not achieve at least partial remission after previous therapies. In these studies, after 12 months of treatment with either a combination of thalidomide and dexamethasone or bortezomib and dexamethasone, 83 out of 95 patients showed significant improvements [20].

Our patient underwent a comprehensive medical evaluation, and the results encouraged us to administer CyBorD (the standard therapy for AL amyloidosis) combined with venetoclax. He received six cycles of this protocol with venetoclax at a dose of 100 mg per day. We also initiated ursodeoxycholic acid based on promising studies demonstrating their successful use in AL amyloidosis patients with severe hepatic involvement, marked by elevated alkaline phosphatase and γ-glutamyl transferase levels [21]. Our patient achieved complete remission with significant improvement in his liver test results. An autologous stem cell transplantation was successfully performed. Venetoclax maintenance therapy was reinitiated on the 100th post-transplant day. According to ongoing regular follow-up examinations, our patient is in a state of complete hematological remission, and his dFLC and level of serum-free light chains have almost normalized.

Our case report confirms the notion that venetoclax can be highly effective in selected patients, not only as a salvage or second-line treatment but also as a part of a first-line induction therapy in t(11;14) AL amyloidosis. Furthermore, venetoclax maintenance can also be useful in these cases. Our positive, though limited experiences with venetoclax emphasize the importance of conducting larger studies to explore its further potential.

## Figures and Tables

**Figure 1 curroncol-32-00063-f001:**
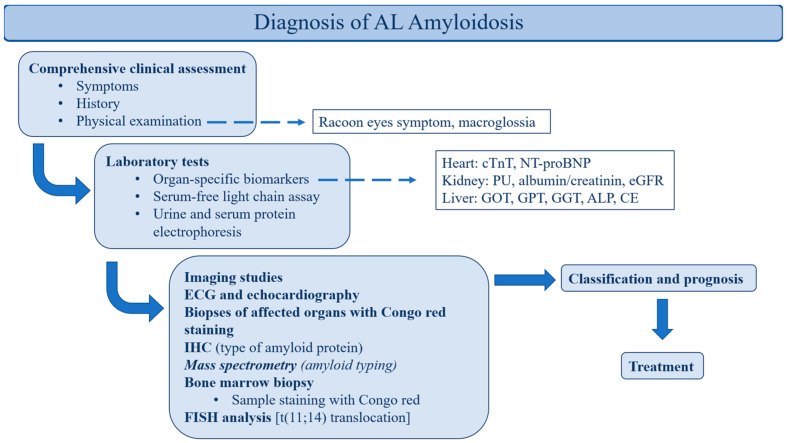
Diagnosis of AL amyloidosis. cTnT: cardiac troponin T; NT-proBNP: N-terminal portion of pro-brain natriuretic; peptide type B; PU: proteinuria; eGFR: estimated glomerular filtration rate; GOT: glutamate oxaloacetate transaminase; GPT: glutamate pyruvate transaminase; GGT: gamma-glutamyl transferase; ALP: alkaline phosphatase; CE: cholinesterase, ECG: electrocardiogram; IHC: immunohistochemistry; FISH: fluorescence in situ hybridization.

**Figure 2 curroncol-32-00063-f002:**
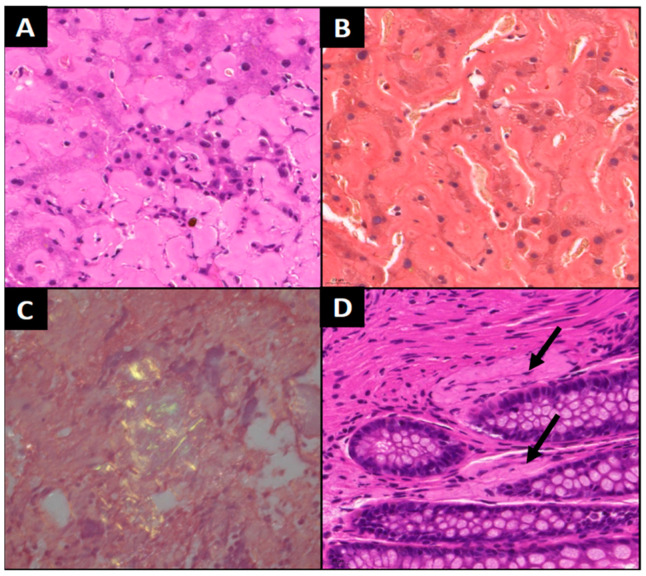
Histological analysis. Eosinophil extracellular deposition and plasmacytoid cells in the bone marrow ((**A**)—H&E 400×); extracellular deposition with congophilia ((**B**)—Congo red, 400×) and double refraction of polarized light ((**C**)—Congo red, 400×). Intravascular amyloid deposits in coecum samples—arrows ((**D**)—H&E, 400×).

**Figure 3 curroncol-32-00063-f003:**
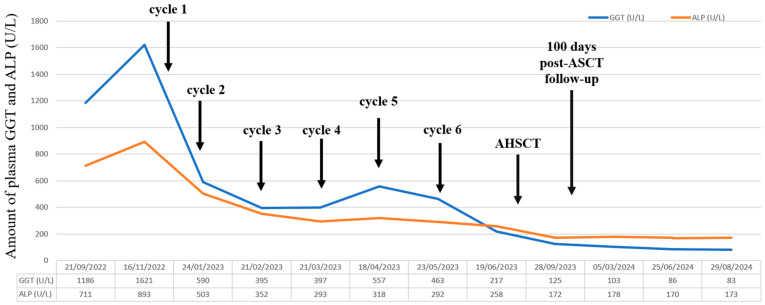
Gamma-glutamyl transferase (GGT) and alkaline phosphatase (ALP) levels during induction and maintenance treatment.

**Figure 4 curroncol-32-00063-f004:**
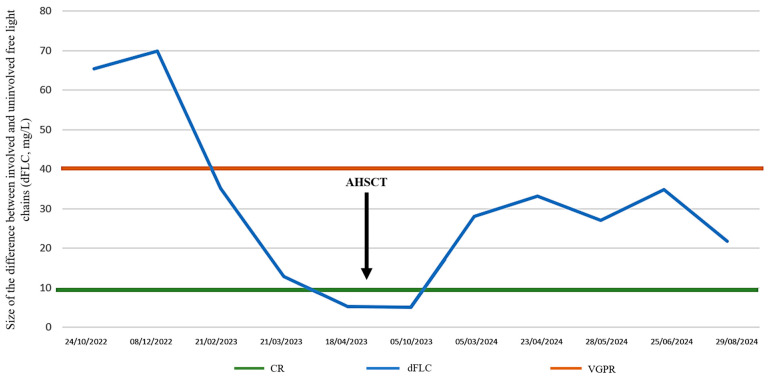
Difference between involved and uninvolved free light chains (dFLC) during treatment before and after autologous stem cell transplantation (AHSCT, 20 June 2023). VGPR is defined as dFLC under 40 mg/L, while CR is defined as dFLC under 10 mg/L. CR: complete remission; VGPR: very good partial remission; dFLC: difference between involved and uninvolved free light shains.

## Data Availability

The original contributions presented in this study are included in the article. Further inquiries can be directed to the corresponding author.

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
