# Peer review of "Venetoclax Plus CyBorD Induction Therapy and Venetoclax Maintenance Treatment for Immunoglobulin Light Chain Amyloidosis with t(11;14) Translocation"

_curroncol, 2025, doi:10.3390/curroncol32020063_

Round 1
Reviewer 1 Report
Comments and Suggestions for Authors
The english in the text should be improved. U should also comment on the CongoRed stain: quite often is negative! and this should not discourage the quest for assigning the diagnosis
Comments on the Quality of English Languagemust be improved in several sections
Author Response
Dear Reviewer,
Dear Editorial Bord,
Thank you very much for the thorough revision of my manuscript. In the following, I would like to present my answers to your questions as well as the corrections I made based on your suggestions.
Reviewer 1
„The english in the text should be improved. U should also comment on the CongoRed stain: quite often is negative! and this should not discourage the quest for assigning the diagnosis.”
Thank you for your revision! Examination of tissue samples stained with Congo red using flourescence microscopy is an important tool for detecting amyloid deposits, it is crucial in the diagnosis of light chain amyloidosis. However, it is important to know, that the sensitivity of this staining is highly dependent on the origin of the sample. For bone marrow samples, the sensitivity is lower compared to samples from the liver, kidneys, or heart muscle. Therefore, if there is a stron clinical suspicion and the Congo red staining yields a negative result it is recommended to support the suspicion with other diagnostic methods, such as IHC.
These data were added to the text (page 2, line 52-56). K. Bowen, N. Shah, and M. Lewin, “AL-Amyloidosis Presenting with Negative Congo Red Staining in the Setting of High Clinical Suspicion: A Case Report,” Case Rep Nephrol, vol. 2012, pp. 1–4, 2012, doi: 10.1155/2012/593460. was added to References.
We apologize for the quality of English language in the original manuscript. Based on your suggestion we have utilized author services language editing to make the necessary corrections.
Sincerely yours,
Gréta Garami
Corresponding author

Reviewer 2 Report
Comments and Suggestions for Authors
This is a single case report of the addition of venetoclax to Cybord chemotherapy for amyloidosis. The case is well-written and documented in detail.
My main criticism is that both the title and abstract are somewhat misleading. When I first read them I thought the patient received venetoclax monotherapy.
Please add the term CyBorD to both the title and abstract
Author Response
Dear Reviewer,
Dear Editorial Bord,
Thank you very much for the thorough revision of my manuscript. In the following, I would like to present my answers to your questions as well as the corrections I made based on your suggestions.
Reviewer 2
„My main criticism is that both the title and abstract are somewhat misleading. When I first read them I thought the patient received venetoclax monotherapy. Please add the term CyBorD to both the title and abstract.”
Thank you for your suggestion about the title and the abstract. The missing information was added to the title, and also, the abstract was supplemented with this data on the front page and the line 24-25.
We apologize for the quality of English language in the original manuscript. Based on your suggestion we have utilized author services language editing to make the necessary corrections.
Sincerely yours,
Gréta Garami
Corresponding author

Reviewer 3 Report
Comments and Suggestions for Authors
Although not usually of sufficient relevance to merit publication, single case reports can have value in describing innovative approaches, particularly where that approach has been informed by particular characteristic(s) of the tumour cells. In the 61 year old male patient described here, that was the presence of t(11;14) which, in turn predicted over expression of the apoptotic inhibitor bcl-2. Hence, the choice of venetoclax, an inhibitor of bcl-2, to supplement the standard drug regimen in the authors' department.
In addition to the recorded clinical and laboratory values, were there any other blood parameters with results before and after treatments that may be relevant? For example, was there any evidence of renal impairment and, if so, was this improved following the addition of venetoclax? Was the presumed cardiac insufficiency lessened following treatment? Can the authors' give a succinct recommended investigative protocol for such cases, to improve early diagnosis and risk stratification? (Possibly a flow chart) Would an international register of these uncommon forms of plasma cell dyscrasia, including treatment strategies, help to overcome the lack of therapeutic trials to inform their management, due to this relative rarity (or possible failure to correctly identify at presentation)?
Comments on the Quality of English Language
The manuscript presentation requires considerable editing, particularly with regard to grammatical and punctuation issues. The authors may benefit from input by someone whose first language is English. Also, during this editing the authors should check for repetition, and be more succinct where possible, thus allowing for some additional information to be included, such as suggested above, without unduly lengthening the manuscript.
Author Response
Dear Reviewer,
Dear Editorial Bord,
Thank you very much for the thorough revision of my manuscript. In the following, I would like to present my answers to your questions as well as the corrections I made based on your suggestions.
Reviewer 3
“In addition to the recorded clinical and laboratory values, were there any other blood parameters with results before and after treatments that may be relevant? For example, was there any evidence of renal impairment and, if so, was this improved following the addition of venetoclax?”
Thank you for your thoughtful and detailed questions! In our case report we aimed to include the most relevant laboratory values related to our patient’s condition. Obviously, there are additional parameters according to the affected organs that we should examine before and after treatments.
For renal involvement, important parameters to monitor would include eGFR, proteinuria and the urine albumin-to-creatinine ratio. In our case the patient did not have renal impairment, therefore these tests were not applicable. For further information on renal AL amyloidosis, we refer to relevant literature, such as the article by Shafqat, A. et al., Renal AL Amyloidosis: Updates on Diagnosis, Staging, and Management. (J. Clin. Med. 2024, 13, 1744).
“Was the presumed cardiac insufficiency lessened following treatment?”
At diagnosis we referred our patient for cardiac evaluation as well. Laboratory results and transthoracic echocardiography revealed cardiac involvement: diastolic dysfunction, left ventricular enlargement, elevated cTnT (40,85 ng/L) and NT-proBNP (409 ng/L) levels. After the 100th day post ASCT we performed a follow-up echocardiography which showed improvement in the thickness of the left atrium wall, it decreased from 47 mm to 38 mm. Laboratory results also showed reduction in NT-proBNP to 204.2 ng/L and cTnT to 18.47 ng/L, compaired to the initial levels of 409 ng/L and 40,85 ng/L respectively. We initiate angiotensin converting enzyme inhibitor for our patient due to the mild cardiac failure.
“Can the authors' give a succinct recommended investigative protocol for such cases, to improve early diagnosis and risk stratification? (Possibly a flow chart)”
We appreciate your suggestion regarding the development of a succinct investigative protocol. Based on our experience and the current literature we have created a flow chart about the main parts in diagnosis of AL amyloidosis. This flow chart has been added to the manuscript as Figure 1 (page 2, line 61-70), and we also added data in the text (page 2, line 60-61).
“Would an international register of these uncommon forms of plasma cell dyscrasia, including treatment strategies, help to overcome the lack of therapeutic trials to inform their management, due to this relative rarity (or possible failure to correctly identify at presentation)?”
Absolutely, international registers for rare forms of plasma cell dyscrasia, such as AL amyloidosis, could play an important role in addressing the challenges associated with the rarity of the condition. These registers would not only help improve the diagnosis, but also provide valuable insights into treatment strategies, offering the opportunity to select the most effective treatment options for our patient
We apologize for the quality of English language in the original manuscript. Based on your suggestion we have utilized author services language editing to make the necessary corrections.
Sincerely yours,
Gréta Garami
Corresponding author

Round 2
Reviewer 3 Report
Comments and Suggestions for Authors
This version of the manuscript is much improved, A few, mainly pedantic, points remain but these do not affect the reader's understanding. (Examples include t(11;14) translocation - analogous to saying reversed backwards!! - no need to include translocation, t(11;14) will suffice. Also, it wasn't always clear if a BM (trephine) biopsy, or a BM aspirate, or both, were taken)